# Pain Neuroscience Education and Motor Control Exercises versus Core Stability Exercises on Pain, Disability, and Balance in Women with Chronic Low Back Pain

**DOI:** 10.3390/ijerph19052694

**Published:** 2022-02-25

**Authors:** Sahar Modares Gorji, Hadi Mohammadi Nia Samakosh, Peter Watt, Paulo Henrique Marchetti, Rafael Oliveira

**Affiliations:** 1Department of Biomechanics and Corrective Exercises and Sports Injuries, University of ARAK, Arak 38156879, Iran; 2Department of Biomechanics and Corrective Exercises and Sports Injuries, University of Kharazmi, Tehran 15719-14911, Iran; shahramsamakosh92@yahoo.com; 3Environmental Extremes Lab, Sport and Exercise Science and Medicine Research and Enterprise Group, University of Brighton, Eastbourne, East Sussex, Brighton BN2 4AT, UK; p.watt@brighton.ac.uk; 4Department of Kinesiology, California State University, Northridge, CA 91330, USA; paulo.machetti@csun.edu; 5Sports Science School of Rio Maior–Polytechnic Institute of Santarém, 2140-413 Rio Maior, Portugal; rafaeloliveira@esdrm.ipsantarem.pt; 6Research Center in Sport Sciences, Health Sciences and Human Development, 5001-801 Vila Real, Portugal; 7Life Quality Research Centre, 2140-413 Rio Maior, Portugal

**Keywords:** therapeutic intervention, non-pharmacologic treatment, non-surgical treatment, female, agility, balance

## Abstract

Background: Several interventions have been used to relieve chronic low back pain (CLBP). This study aimed to compare the effects of pain neuroscience education (PNE) followed by motor control exercises (MCEs) with core stability training (CST) on pain, disability, and balance in women with CLBP. Methods: Thirty-seven women with CLBP were randomly divided into two groups of PNE/MCE (n = 18, 55.2 ± 2.6 years) or CST (n = 19, 54.6 ± 2.4 years). Eight weeks of PNE/MCE or CST were prescribed for each group, independently. Pain intensity (VAS scale), disability (Roland Morris Disability Questionnaire), unipodal static balance, and dynamic balance (time up and go test) were measured at the beginning and 8 weeks after the intervention. Two-way mixed ANOVA was used to analyze the results with alpha of 5%. Results: After 8 weeks, there was a significant difference in VAS scale between groups (*p* = 0.024), with both PNE/MCE and CST showing 58% and 42% reductions, respectively. There were no differences for all other variables between groups. Regarding pre- to post-comparisons, both groups showed improvements in all dependent variables (*p* < 0.001). Conclusion: The treatment with PNE/MCE was more effective in improving pain disability and unipodal static and dynamic balance than treatment with CST. Even so, both treatments were shown to be valid and safe in improving all dependent variables analyzed in women with CLBP.

## 1. Introduction

Chronic low back pain (CLBP) is considered a frequent condition in women that has increased significantly worldwide, but few studies have been conducted to explain it [1,2]. Low back pain (LBP) is associated with disorders and disabilities in balance [3,4] and movement [5]. Several non-surgical treatments have been offered to help and decrease LBP, such as joint manipulation [2], acupuncture [6], traditional therapeutic exercises [7,8], and medication [9,10]. However, the small effects of these treatments indicated that effective treatment methodologies are needed to achieve a significant therapeutic effect.

Moreover, several methodologies such as strength, endurance, and core training programs available for treating CLBP seem to be weak because they ignore pain recognition, behavioral aspects, and knowledge of pain physiology [1].

Meanwhile, spinal stabilization exercises for CLBP have received some research attention and are regarded to be effective, through improved core stability of the central axis of the role of these muscles in this area during limb use and movement [11]. In fact, it was found that a core stability training (CST) program significantly increased the multifidus muscle cross sectional area in women without and with CLBP [12]. This type of exercise focuses on the ability of the spine to stabilize in different positions and emphasize smaller, deeper, and posterior muscles. By increasing the endurance of these muscles, they retrain and maintain the correct physical condition to stabilize the spine to improve pain, performance, and balance [13]. Moreover, the weakness of lumbar multifidus muscles can impair the function of the spine during both dynamic movements and static positions. Thus, training protocols for LBP have aimed to reduce it through lumbar stability exercises that target multifidus muscles [14,15].

In addition, more knowledge of spinal motor dysfunction in patients with CLBP has emerged [16,17]. Patients with CLBP that present reduced motor control usually have difficulties to control postures and movements [16,18].Therefore, addressing pain recognition and a better spinal muscle function to improve spinal motor control could be more effective [19,20,21].

A methodology that included pain neuroscience education (PNE) followed by motor control exercises (MCEs) has recently been used [22,23]. This methodology aims to educate about the mechanism of CLBP (central pain, central sensitivity, and cognitive–sensory mechanisms of pain) followed by individual MCEs to help patients develop activities that they do with fear and hesitation [24].

In this sense, a study showed that a PNE program combined with motor control education with cognitive goals presented decreases in pain and disability and an increase in spinal function compared to physiotherapy intervention in patients with CBP. Moreover, a recent systematic review and meta-analysis showed that exercise training programs obtained identical clinical effects when compared with physiotherapy interventions [25].

Given the scarce knowledge on how effective the previous interventions are, the scarce information on what kind of exercises are best to improve LBP [25,26,27,28,29,30], and that supervised exercise therapies have been suggested as primary methodology for CLBP in recent years [11], more studies are needed to better combat CLBP. For instance, a recent meta-analysis regarding CST versus general exercise training programs found that CST is more effective than general exercise in decreasing pain and improving physical function in patients with chronic LBP [31]. However, PNE/MCE was not taken in consideration. Based on the previous information, this study aimed to compare two interventions (PNE/MCE and CST) on pain, disability, and balance in women with CLBP. We hypothesized that participants that underwent PNE/MCE would present a better improvement in pain intensity, disability, and balance compared to CST.

## 2. Materials and Methods

### 2.1. Participants

Sixty-five participants aged between 50 and 60 years with CLBP were enrolled in this study. CLBP was diagnosed by a specialist physician by the cross straight leg raise test. The participants were purposefully selected, and only those registered for physiotherapy treatment through brochures displayed in physiotherapy clinics were enrolled by physiotherapists. The sample size was calculated using G*Power as in a previous study [22] with the effect size difference of 0.80, alpha value of 0.05 (two-way test), and a power (1 − 𝛽) of 0.80. A total sample size of 42 (21 patients in each group) was required.

Thus, the following inclusion criteria were applied: (1) native Persian-speaking women between 50 and 60 years old and (2) primary complaints of low back pain (more than 3 months, usually between the lower ribs and the creased part of the buttocks without nonspecific pathoanatomical cause) diagnosed by an experienced physiotherapist and completion of the consent form. The exclusion criteria were (1) any history of surgery on the spine, (2) pathological records (disease) of the spine, (3) orthopedic and neurological injuries, (4) unwillingness to continue to participate in the exercises, (5) failed any exercise session (2 consecutive or 3 non-consecutive sessions, n = 3), and (6) absence in the post-test stage (n = 2) [24]. Additionally, participants were excluded from the study if they engaged in parallel therapy, in special work activities, or regular exercise. Prior to randomization, socio-descriptive data and primary clinical outcome variables were collected from all participants and recorded by an uninformed evaluator. Then, forty-two participants were randomly assigned to one of two treatment groups with a ratio of 1:1 as follows: PNE/MCE group (n = 21) and CST group (n = 21). Figure 1 describes the participants’ enrollment in the study. Randomization was performed by a blinded researcher who had no knowledge of the study. Randomization was performed with naming codes specific to each person (up to 42), which had already been sealed: the blinded person was asked to place 21 papers in both balls. The participants were informed of the risks and benefits of the study prior to any data collection, and then, participants signed the informed consent form to be examined according to the Declaration of Helsinki, and they declared their voluntary participation. The present study was registered and approved by the local ethics committee of ARAK (1399.12.2.5).

### 2.2. Procedures

The present research design was a two-group study with exercise intervention in the experimental groups (PNE/MCE and CST groups) without a control group with pre- to post-test. A random method to allocate participants in the groups was used with blinding in the experimental group. The following evaluation was applied to all participants:

Visual Analog Scale (VAS): A visual analog scale was used to measure the severity of pain in the subjects. The scale is a 10 cm horizontal strip starting from 0 (no pain) to 10 (the most severe pain possible). This scale is one of the most reliable quantitative scales widely used in research [32].

Disability: The Persian version of the Roland–Morris Disability Questionnaire (RMDQ) was used to assess disability by LBP [33]. The RMDQ is a 24-item questionnaire reported by the patient. Among the questions related to disability is LBP-induced pain. Items are scored from 0 to 24 for a total RMQ score: 0 and 1 are the scores of blank and confirmed items, respectively. Higher scores indicate higher levels of pain-induced disability [33].

Unipodal static balance (USB): The test of standing on the dominant foot on a square piece of wood (40 cm × 40 cm) was used to measure the static balance of the subjects. The procedure is performed in a manner that the subject stands barefoot in such a way that the superior foot is on the ground and the non-superior foot is above the ground, and the hands are placed on the waist on the crown of the pelvis. The length of time a person can maintain this position (in seconds) is counted as their score. Time recording stops when the support foot moves, the free foot touches the ground, or the hand is detached from the waist [33].

Dynamic balance: The “Timed Up and Go (TUG) test” was used as a rapid way to determine balance problems affecting motor skills in the daily lives [34]. The TUG test consists of three steps: (1) getting up from the chair, (2) walking 3 m away, (3) turning around a cone, walking back, and sitting on the chair. The subject raises from a chair (without a handle) without using their hands. The time is defined from the moment the examiner commands the subject to get up to the time the subject sits back on the chair in the correct position (leaning on the back of the chair). The subject has to perform this test in the shortest possible time [35].

### 2.3. Interventions

Core stability training: After familiarization with all core stability exercises, all participants performed the CST for eight weeks (3 sessions per week, 45–60 min each session) supervised by a physiotherapist. The CST in each session was composed of a warm-up (walk for 5 min, stretching exercises for 5 min, the main intervention for 30 min (in the initial weeks) to 45 min (in the final weeks), and cool down for 5 min. Progress in exercises was based on the principles of overload and gradual increase in the volume of exercises; progress was in the duration or repetition of each exercise [36]. Table 1 presents all exercises performed across the eight weeks.

Pain neuroscience education/motor control exercises: Three PNE sessions took 30–60 min and were conducted by a physical therapist. The purpose of PNE was to control patients’ negative perceptions of recurring pain [22,37]. These beliefs may be imposed on patients by potentially useless diagnostic, prognostic, or therapeutic conclusions. During the PNE sessions, pain information was given to avoid fear beliefs and behaviors, thus providing keywords at this stage to promote self-efficacy using verbal instructions, charts, and freehand drawings [22,37] (Figure 2).

The MCEs were similar to Malfliet et al. [22]. This phase consists of a proprioception, coordination, and sensorimotor control training program based on the principles and ideas of researchers and clinicians such as Sahrmann [38], Comerford and Mottram [39], and Richardson and Jull [40]. Participants underwent 16 sessions of MCEs (twice a week) for eight weeks. In the first session, participants were examined individually by a physical therapist (who performed the PNE exercises). The exercises prescribed were based on the participant’s tolerance/ability. MCEs were performed separately and supervised by a physical therapist, and it was composed of three parts with specific criteria for each patient. MCEs included sensory–motor control training with the facilitation of the proprioceptive system and optimization of coordinated muscle patterns. Patients were instructed to contract the deep muscles of their spine (e.g., transversus abdominis and multifidus) that are separate from the superficial muscles. The exercises progressed with the addition of specific exercises for the diaphragm and pelvic floor muscles. The exercise progression was ensured to the extent that each individual could maintain 10 s per repetition of isolated contractions. Then, another part of progression was applied [22]. In the second part, additional loads were added on the spine by performing different patterns of lower limb (shoulder bridge) and trunk movement (lumbar extension) while using the deep and superficial muscles of the spine. In the third part, functional exercise was added, focusing on static followed by dynamic functional exercises aiming at improving coordination, alignment, and stability of the spine. During all sessions, the posture, movement technique, and breathing status were assessed. In order to adapt the exercises to the daily situations of people, the progress of the exercise was done during physical activities and daily activities or sports in stressful psychological conditions [22]. Table 2 presents MCE training.

### 2.4. Statistical Analysis

The Shapiro–Wilk test and Levene’s tests were conducted to confirm the normality and homogeneity of the dependent variables, respectively. After both were confirmed, a 2 × 2 ANOVA (treatment group × time) with Bonferroni correction post hoc was conducted with a mixed model analyses design. For each dependent variable, mean ± standard deviation (SD) was used. The percentage of change was calculated and compared with the baseline. A *p*-value < 0.05 was used to determine statistical significance, while Partial Eta Squared (*η*p^2^) values were calculated as effect size (ES), which were considered as 0.2 = small effect, 0.5 = moderate effect, and 0.8 = large effect based on the study of Cohen [41]. All data analyses were conducted on SPSS software version 26 (IBM Corp., Armonk, NY, USA).

## 3. Results

There was no significant difference between the two groups in the pre-test stage in terms of demographic and clinical variables (*p* > 0.05). The demographic and clinical variables of the participants are presented in Table 2. Five patients did not participate in the post-test for personal reasons (PNE/MCE = 3 and CSE = 2), with 88% retention in the study (37 of 42) (Figure 2).

Table 3 shows the effect of 8 weeks of intervention and the comparisons between time and groups. Overall, there were significant improvement between baseline and after 8 weeks for both groups. VAS presented significant differences between groups, while no significant differences were found in the dependent variables of RMDQ, USB, and dynamic balance between groups.

There was a difference in the mean score of pain (−0.78) (PNE/MCE, ES = 2.14, CSE, ES = 3.43) and disability (0.79) (PNE/MCE, ES = 5.13, CSE, ES = 2.62) (RMDQ) (better performance in the PNE/MCE group) in the post-tests. Additionally, the mean difference in the PNE/MCE group and the core stability group was 0.79 (PNE/MCE, ES = −4.87, CSE, ES = −4.12) in the USB and −0.62 (PNE/MCE, ES = 4.49, CSE, ES = 4.46) in the dynamic balance variable (better performance in the PNE/MCE group) (Table 4; Figure 3 and Figure 4).

## 4. Discussion

The aim of this study was to compare two interventions, PNE/MCE and CST, on pain, disability, and balance in women with CLBP. The results showed a significant effect of both interventions after eight weeks. Additionally, the comparison of the two exercise programs showed a greater effect of motor control with cognitive purpose compared to the core stability exercises on the variables of pain and disability (VAS and RMDQ, respectively). A possible justification could be associated with the individualized, PNE treatment employed using cognitive-based education followed by an exercise program as previously suggested [21].

Moreover, the findings of many previous studies are consistent with the present results in terms of core stability and motor control exercises with a cognitive goal [1,24,42,43]. Core stability exercises are designed and performed to improve body trunk muscles with the aim of developing muscle strength and functional coordination [44,45]. On the other hand, CLBP is aggravated by improper activities and causes most patients not to use their back, which leads to the atrophy of trunk muscles, decreased muscle strength and endurance, and stiffness of ligaments and joints, which in turn aggravate the symptoms of CLBP [46].

The results of some studies have shown that the size of type II muscle fibers is reduced in people with LBP [34,47]. Stability exercises with maximal or submaximal effort can reverse the atrophy of type II fibers in multifidus muscle and affect the diameter of the muscle fiber [48]. Regarding the effect of CST on balance, it should be noted that the body’s center of gravity is constantly shifted during dynamic activities. Every voluntary movement disturbs the physical condition. If this perturbation is not anticipated, the balance may be disturbed. The muscles that surround the core play a vital role in center of gravity [49]. A stable proximal region is required to create a better distal motor chain with pre-programmed muscle activations. The preventive contraction of deep trunk muscles and multifidus before limb movement improves trunk stability, which reduces joint forces and abnormal movements in the distal parts [50]. In our study, despite the fact that improvements in USB and TUG were found, the non-significant group x interaction analysis could not support the effectiveness of CST for both tests, which require more studies to confirm the results.

In people with CLBP, the trunk protection response is inadequate due to a delay in predictive activation and a reduction in relative muscle thickness [51]. The disability of trunk muscles can contribute to an instable pelvic floor, which may consequently increase the risk of injury. In this regard, Hooper et al. reported that Y balance test performance (anterior, posterolateral, and posteromedial directions) decreased in patients with CLBP [52].

When compared with static balance, dynamic balance requires the body to balance during the transition from dynamic to static status, which is difficult [53] for people with CLBP who struggle to adapt to changing conditions [54]. Core stability exercises improve torso stability, pelvic floor stability, the simultaneous contraction of the abdominal muscles, multifidus, and improved motor performance of the spine [55], and reduce the shear force applied to the waist [56], thereby improving the movements of the lower limbs in static positions and dynamic movements, which corroborates with the present study. Exercises such as the simultaneous contraction of abdominal muscles, multifidus, diaphragm, and pelvic floor along with exercises related to improving performance and controlling limb function in the program of motor control exercise with cognitive purpose can be an effective factor in improving static and dynamic balance [57]. In general, the muscles of the central region are the postural muscles. During whole-body exercises, an important role in the stability of the trunk and control of the body position was performed, which neutralize the muscle imbalances to maintain the body position, thus reducing disability [58].

With regard to the effect of PNE/MCE with a cognitive goal, a growing body of evidence shows that brain abnormality changes in the structure and function of the brain and excessive brain sensitization have major importance in patients with CLBP [1,24]. Central sensitization includes altered sensory processing in the brain [1]. In addition, brain plasticity causes pain and fatigue, which can lead to disability even without actual tissue damage or pain [24]. In this regard, PNE intervention presents significant improvements to combat these issues [22]. In the present study, central sensitization was not assessed. Even so, it has been speculated that educating patients about how to control pain and their feeling perceptions will help in the excessive reduction in central nervous system sensitization and pain relief [42,43,59]. This factor, along with the practice of exercises related to the muscles of the central region, may have been a factor in the greater effectiveness of this exercise program than the exercises of core stability on pain and disability. Physical feedback and cognitive biofeedback along with the implementation of exercise programs have been suggested in some studies to obtain better results in the elderly [60].

The addition of PNE to MCE seems to help patients by reducing pain and disability by activating proprioception, coordination, and sensory–motor control of the spine [42,43,59]. Malfliet et al. also found that PNE/MCE with a cognitive goal is more effective than the current most common physiotherapy exercises to reduce trunk pain and disability [22]. Furthermore, Fletcher et al. found that patients with more pain knowledge showed lower levels of pain fear and disabilities [61]. Moreover, the USB and TUG tests showed improvements after 8 weeks of training, but the non-significant group x interaction analysis could not support the absolute effectiveness of PNE/MCE.

This study presents limitations, namely: (a) only eight weeks were evaluated, which suggests more longitudinal studies are needed to access long-term effects of the interventions; (b) central sensitization was not accessed because there is no translated Persian version; (c) brain imaging techniques were not used, which could be relevant for future studies to understand brain communication during CLBP recovery and during the application of the different training methodologies; (d) it was not possible to apply the ideal division for all groups, which would include five different groups: (1) MCE; (2) PNE/MCE; (3) CST; (4) PNE/CST, and (5) the control group. For those reasons, we recommend such an approach for future studies.

## 5. Conclusions

According to the percentage of changes, PNE/MCE was shown to provide more improvements than CST in relieving pain and to some extent in the RMDQ disability than in the CST group. This is probably due to the focus of neuroscience with motor control exercises on the correction of incorrect postures during daily activities. In addition, PNE/MCE were shown to possibly be effective in improving static and dynamic balance, although further studies would be needed to substantiate this. Even so, both PNE/MCE and CST protocols improved all variables analyzed (pain, disability, and static and dynamic balance). Therefore, both training programs seem to be recommended for women with CLBP.

## Figures and Tables

**Figure 1 ijerph-19-02694-f001:**
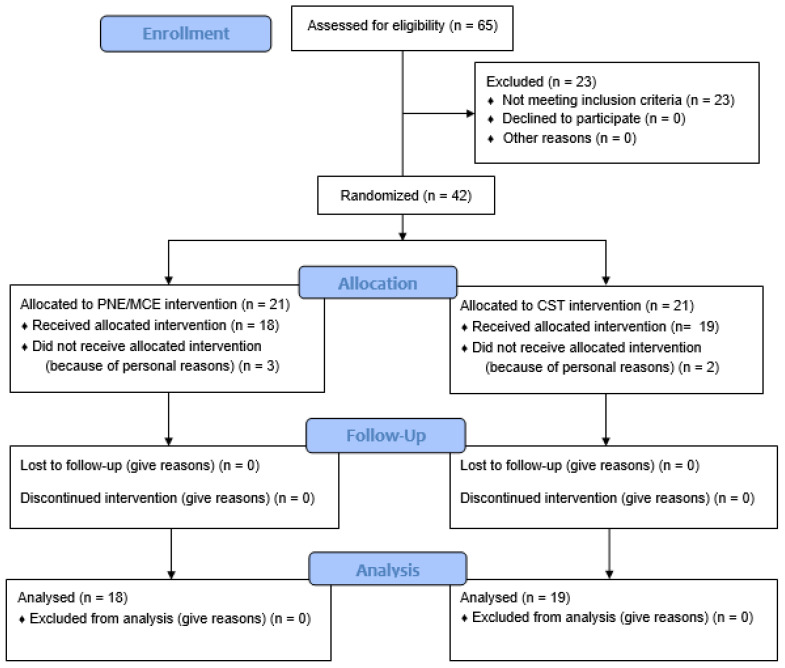
CONSORT flow diagram of the study.

**Figure 2 ijerph-19-02694-f002:**
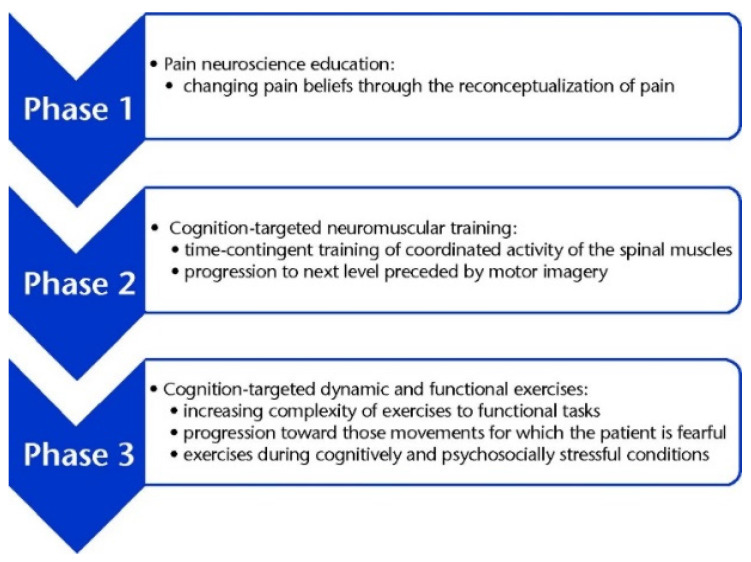
Parts related to pain neuroscience.

**Figure 3 ijerph-19-02694-f003:**
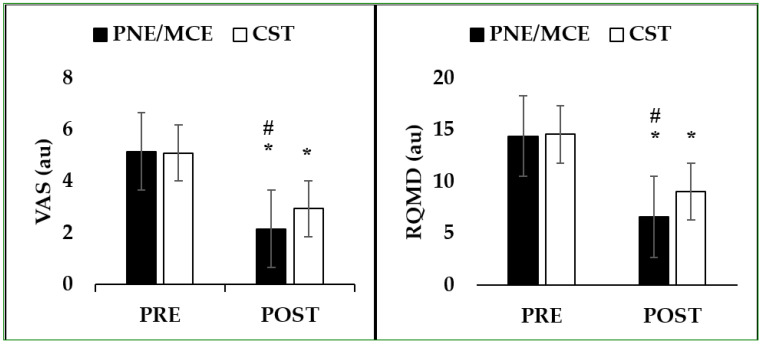
Pre- to post-test of VAS scores (range from 0, “no pain” to 10, “high pain”) and Roland–Morris Disability Questionnaire (RMDQ), scores range from 0, “no pain-related disability,” to 24, “high pain-related disability”. * denotes difference between pre to post test (*p* < 0.05). # denotes difference between PNE/MCE versus CST (*p* < 0.05).

**Figure 4 ijerph-19-02694-f004:**
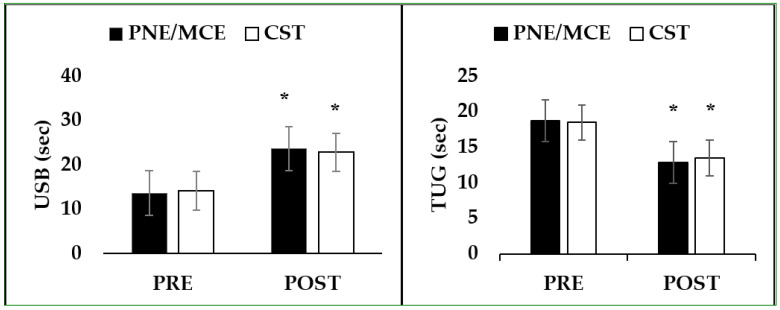
Pre- to post-test of USB and Timed Up and Go test (TUG). * denotes difference between pre to post test (*p* < 0.05).

**Table 1 ijerph-19-02694-t001:** Core stability exercises in one session.

Exercise	Sets and Repetitions per Week							
	1–2 Weeks		3–4 Weeks		5–6 Weeks		7–8 Weeks	
	Set	R/S	Set	R/S	Set	R/S	Set	R/S
Stomach abduction (static contraction of the abdominal muscles) (S)	3	20	3	25	4	30	5	30
Four-legged position with raising the opposite arm and leg (R)	3	20	3	25	4	30	5	30
Single leg adjusted side bridge for each side of the body (R)	3	10	3	15	4	15	5	15
Supine on a Swiss ball with static contraction of the abdominal muscles (S)	3	10	3	15	4	15	5	15
Raise the opposite arm and leg on the Swiss ball (R)	3	10	3	15	4	15	5	15
Standing on one leg with the knee flexed (S)	3	10	3	15	4	15	5	15
Standing on two legs with balance sandals in an anatomical position or with eyes closed (S)	3	8	3	12	4	15	5	15
Standing with feet on the wobble board (S)	3	8	3	12	4	15	5	15
Standing and walking with balance sandals on two legs	3	8	3	12	4	15	5	15
Stride using balance sandals	3	8	3	12	4	15	5	15
Standing and walking with balance sandals and with the knee flexed at feet	3	8	3	12	4	15	5	15

Legend: S = second, R = repetition.

**Table 2 ijerph-19-02694-t002:** MCE training for 8 weeks.

Phase	Weeks	Set	R/S	Exercises
A	1–2 Weeks	3	10	Pelvic tilt, double leg stance, bridge, cat and cow exercise
	3–4 Weeks	3	10	Single leg stance, single leg bridge, cobra with hands off floor, quadruped trunk rotation
B	5–6 Weeks	4	15	Single leg stance eyes closed, flexion and extension of the back without weights, straight leg raise, walking on stable board
	7–8 Weeks	5	15	Forward bending, flexion and extension of the back with weights on unstable board, walking on unstable board, cross straight leg raise, eccentric squat

S = Second, R = Repetition.

**Table 3 ijerph-19-02694-t003:** Demographic data and baseline values of patients with chronic low back pain.

Characteristic	PNE/MCE (n = 18)	CST (n = 19)	*p*-Value
Age (years)	55.16 ± 2.61	54.63 ± 2.38	0.52
Body weight (kg)	1.63 ± 0.06	63.36 ± 4.75	0.55
Body height (m)	63.72 ± 5.47	1.65 ± 0.04	0.83
Body Mass Index (kg/m^2^)	23.56 ± 1.61	23.08 ± 0.89	0.47
Low back pain duration, months	4.00 ± 076	3.89 ± 0.73	0.67

Legend: CST, core stability training; PNE/MCE, pain neuroscience education followed by motor control exercises.

**Table 4 ijerph-19-02694-t004:** Pre- and post-test comparisons of the outcomes assessed in the study and between groups.

Dependent Variables	Group	Baseline Mean ± SD	Post 8 Weeks Mean ± SD	Δ Pre–Post	Main Effect: Time			Main Effect: Group			Interaction: Time × Group		
					F	*p*	*η*p^2^	F	*p*	*η*p^2^	F	*p*	*η*p^2^
VAS (au)	PNE/MCE	5.16 ± 0.70	2.16 ± 0.072	↓58%	563.34	<0.001 *	0.971	6.16	0.024 *	0.266	4.21	0.05 *	0.198
	CST	5.10 ± 0.80	2.94 ± 0.91	↓42%									
RMDQ (au)	PNE/MCE	14.38 ± 1.94	6.61 ± 0.92	↓54%	567.06	<0.001 *	0.971	3.45	0.08	0.169	13.69	0.002 *	0.446
	CST	14.57 ± 2.77	9.05 ± 3.55	↓37%									
USB (sec)	PNE/MCE	13.62 ± 1.37	23.59 ± 2.05	↑73%	712.9	<0.001 *	0.977	0.031	0.863	0.002	3.71	0.071	0.179
	CST	14.13 ± 2.16	22.80 ± 1.69	↑61%									
TUG (sec)	PNE/MCE	18.70 ± 0.74	12.84 ± 1.26	↑31%	654.38	<0.001 *	0.975	0.262	0.615	0.015	2.98	0.102	0.149
	CST	18.46 ± 0.96	13.46 ± 1.12	↑27%									

Legend: PNE/MCE, pain neuroscience education followed by motor control exercise; CST, core stability training; Δ, percent change (↓decrease, ↑increase); *η*p^2^, partial eta squared (effect size). VAS: scores range from 0 (“no pain”) to 10 (“high pain”); RMDQ, Roland–Morris Disability Questionnaire: scores range from 0 (“no pain-related disability”) to 24 (“high pain-related disability); au, arbitrary units; STB, unipodal static balance test; TUG, Timed Up and Go Test; sec, second. * significant differences (*p* ≤ 0.05).

## Data Availability

The datasets used or analyzed during the current study are available from the corresponding author on reasonable request.

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
