# Peer review of "Pain Neuroscience Education and Motor Control Exercises versus Core Stability Exercises on Pain, Disability, and Balance in Women with Chronic Low Back Pain"

_ijerph, 2022, doi:10.3390/ijerph19052694_

Round 1

Reviewer 1 Report

General comments

The authors addressed the Chronic Low Back Pain topic and described the most used exercise therapies.

Studies suggest that Core Stability Training is more effective than general exercise to decrease pain and improve physical function in patients with Chronic Low Back Pain.

However, the methodology that included Pain Neuroscience Education followed by Motor Control Exercises was not taken into consideration from the scientific literature. Therefore, the authors aimed to compare the effects of the Pain Neuroscience Education followed by Motor Control Exercises with the Core Stability Training on pain, disability, and balance in women with Chronic Low Back Pain.

The topic is interesting but the manuscript needs some revisions to be improved.

Specific comments

Abstract

In the Background, the aim of the study is missing.

Replace with: "Several  interventions  have  been  used  to  relieve  Chronic Low  Back  Pain (CLBP). This study aimed to compare the effects of the Pain Neuroscience Education (PNE) followed by Motor Control Exercises (MCE) with Core Stability Training (CST) on pain, disability, and balance in women with Chronic Low Back Pain."

Material and Methods

Line 97: Replace with "The sample size was calculated ... as in previous study [16] with the effect ...

Procedures

Line 138: Replace with "Unipodal Static Balance (USB):"

Statistical analysis

Post hoc tests used after group x time interaction was statistically significant are not specified. Please describe the post hoc tests used.

Results

Lines 210-219: The data described in the text are the same as those in table 3. Please remove unnecessary repetitions.

Line 214: Replace "SBT" with "USB".

Table 3: Replace "SBT" with "USB".

Legend of Table 3: "SBT" with "USB".

Figure 4: Remove "Single Leg Stand" from the ordinate axis and replace with "USB". In the Legend replace with "Unipodal Static Balance (USB)".

Figue 3 and 4: It would be necessary to indicate in the text the Effect size of the post-hoc tests (which were used?).

Discussion

I am not very convinced that the experimental intervention is effective on USB and TUG. There is no significant group x time interaction and this means that any pre-post variations are not due to the experimental intervention. I recommend specifying this in the discussions.
Also, in lines 313-314 one must be cautious in formulating conclusions. Replace with: "In addition, PNE / MCE have shown that they may also be effective in improving static and dynamic balance, although further studies would be needed to substantiate this. Even so, both ..."

Reviewer 2 Report

The manuscript ID: ijerph-1591498 seems to me well structured and executed. In particular, the objectives are detailed in a clear and defined way, the detection methods and the working scheme are well illustrated. Tables are clear and easy to read. The very schematic statistical evaluation is adequate for the study. Furthermore, the limitations of the study (on the short duration of observation and others) are well reported in lines 300-308.

Finally, the conclusions are rightly interlocutory, given the structure of the study and the limitations present. I believe that the study, while not particularly innovative or extraordinary from a scientific point of view, is a good example of communicating results in the real world and could be useful for improving clinical practice. For these reasons I believe it is a paper that deserves publication.

Reviewer 3 Report

I would like to congratulate the authors for their work on this study. I do think it contributes to the literature, however, I do have some suggestions for the authors below.

Line 41: LBP does not spread, perhaps the authors imply that LBP is increasingly prevalent? It would be appropriate to include updated Prevalence studies on LBP, or CLBP in females

Line 48, please include primary references in support of the stated interventions instead of borrowing from a study that cites support. There is a wealth of primary references for joint manipulation, exercise and acupuncture for CLBP.

Lines 94-95, can the authors share what types of clinical tests were used in the physician assessment of lower back pain?

Line 172 used the incorrect reference, Ref (16) is Malfliet, A et al. Did the authors mean this individual instead of Macedo?.

Line 173 Could the authors provide more specific examples of the MCE exercises that were used in the study. These should be laid out in table format similar to that of the CST exercises. Either as a Supplementary file or table inserted into the manuscript.

Line 286: incorrect word use, perhaps assessed vs accessed? 

Round 2

Reviewer 1 Report

Thank you for your replies!